# Ageing and Inflammation: What Happens in Periodontium?

**DOI:** 10.3390/bioengineering10111274

**Published:** 2023-11-02

**Authors:** Luying Zhu, Zhongyuan Tang, Renjie Hu, Min Gu, Yanqi Yang

**Affiliations:** Faculty of Dentistry, The University of Hong Kong, 34 Hospital Road, Sai Ying Pun, Hong Kong SAR 999077, China; u3008485@connect.hku.hk (L.Z.); tzy337@connect.hku.hk (Z.T.); u3008527@connect.hku.hk (R.H.); drgumin@hku.hk (M.G.)

**Keywords:** ageing, inflammation, periodontitis, inflammaging

## Abstract

Periodontitis is a chronic inflammatory disease with a high incidence and severity in the elderly population, making it a significant public health concern. Ageing is a primary risk factor for the development of periodontitis, exacerbating alveolar bone loss and leading to tooth loss in the geriatric population. Despite extensive research, the precise molecular mechanisms underlying the relationship between ageing and periodontitis remain elusive. Understanding the intricate mechanisms that connect ageing and inflammation may help reveal new therapeutic targets and provide valuable options to tackle the challenges encountered by the rapidly expanding global ageing population. In this review, we highlight the latest scientific breakthroughs in the pathways by which inflammaging mediates the decline in periodontal function and triggers the onset of periodontitis. We also provide a comprehensive overview of the latest findings and discuss potential avenues for future research in this critical area of investigation.

## 1. Introduction

Periodontitis is a ubiquitous chronic inflammatory disease characterized by the gradual destruction of the periodontal ligament and alveolar bone, leading to periodontal pocket formation and gingival recession [1]. Meanwhile, the presence of periodontal bacteria and viruses increases periodontal destruction [2]. This condition, if left untreated, can lead to the progressive destruction of tooth-supporting tissues, ultimately resulting in tooth loss and dental implant failure, as well as a subsequent decline in oral function and quality of life [3]. Consequently, the pursuit of therapeutic interventions aimed at treating periodontitis and effectively restoring compromised periodontal structures has been a major focus of research in oral medicine in recent years [4]. Ageing is characterized by the deterioration of biological processes in living organisms, resulting in the loss of function [5,6]. Additionally, studies have shown that ageing stands as a significant risk factor for a range of chronic diseases that can limit survival, independence, and prosperity. The older population is particularly vulnerable to various chronic conditions, such as cancers, atherosclerosis, diabetes, autoimmune diseases, and infectious diseases, including periodontitis [5]. Clinical research has demonstrated an increase in both the prevalence and severity of periodontal disease with advancing age, particularly after 30–40 years of age, and further exacerbation in most adults aged over 50 years [7,8,9]. Moderate loss of the alveolar bone and periodontal attachment is common in older individuals [10]. Meanwhile, bacteria worsen the bone resorption around dental implants [11]. However, the exact contribution of ageing to the onset, progression, and remission of periodontitis remains poorly comprehended. An examination of the relationship between ageing and inflammation may pave the way for novel avenues of research in this domain and help identify potential clinical interventions that can reverse or mitigate the impact of ageing on periodontitis. By shedding light on this complex relationship, we may be able to develop effective strategies for improving the oral health and overall quality of life of the ageing population.

## 2. Ageing and Inflammation

Ageing is commonly defined as a time-dependent impairment in physiological functions and is regarded as an inevitable and universal process for all living organisms. Ageing can lead to the dysfunction of multiple cellular and molecular events, resulting in various age-related changes and health issues. One of the major changes associated with ageing is immune response dysregulation, which can lead to the development of a chronic systemic inflammatory state. This, in turn, can hasten the onset and progression of various chronic diseases, ultimately compromising patients’ quality of life and leading to their demise [12]. Ageing is widely recognized as one of the most intricate biological phenomena, governed by a multitude of intricately intertwined mechanisms. In recent years, inflammation has gained recognition as a critical intermediary in various age-related diseases and has been implicated in the onset and advancement of numerous chronic conditions [13].

Inflammation is the immune system’s natural response to injury or infection and is a necessary component of the body’s defense against harmful stimuli. However, chronic inflammation can lead to tissue damage and play a contributory role in the pathogenesis of several diseases, including cardiovascular disease, diabetes, and cancer. As inflammation is one of the key changes that occurs during ageing and also plays a central influence on the development of numerous age-related chronic disorders, it is important to investigate these two processes using an integrative approach [14]. Recent research has highlighted the complex interplay between inflammation and ageing. In 2000, Franceschi et al. [6] first introduced the term ‘inflammaging’ to characterize the gradual elevation of pro-inflammatory status observed with advancing age.

### 2.1. Inflammaging and Immunosenescence

Inflammaging is attributable to a decline in the capability to cope with types of stressors along with a gradual amplification of pro-inflammatory status. It is a major characteristic of the ageing process and is triggered by the sustained burden of antigenic and stress factors [15]. Moreover, inflammaging is thought to arise from immune system dysregulation that occurs with ageing. Recent research has highlighted the complex interplay between the immune system and the stress response, suggesting that the two are equivalent in response to stimuli. Antigens, which are typically considered to be foreign substances that trigger an immune response, can be viewed as a specific variety of stressors [15]. The immune system dysregulation that occurs with ageing can impair the function of both the innate and adaptive immune systems. However, compared with the less well-functioning adaptive immune system, the innate immune system maintains a comparatively higher level of functionality in older individuals [15]. Therefore, inflammaging is related to ‘immunosenescence’, which is another concept associated with immune system and ageing.

The term immunosenescence is used to describe the gradual deterioration in immune system function that occurs with ageing. It is primarily pertinent to adaptive immunity systems, highlighting the diminished and dysregulated responses of T and B lymphocytes in aged individuals, particularly in the context of chronic stressors such as pathogens and infections [16,17]. This decline can impair the immune system’s ability to respond efficiently and to pathogenic challenges, thereby increasing the susceptibility to infections and reducing the efficacy of vaccines. Among the dysregulated pro-inflammatory mediators that contribute to immunosenescence and inflammaging, cytokines, and chemokines are the major culprits.

### 2.2. Cytokines, Chemokines and Pathways in Ageing and Inflammation

The levels of interleukin (IL)-1, IL-6, tumor necrosis factor-α (TNF-α), and their receptors are not only up-regulated in aged tissues and cells, but also elevated in inflammatory diseases [18]. Age-related chronic inflammatory changes are closely linked to several crucial cellular signaling pathways [19,20]. Table 1 presents the pro-inflammatory cytokines, chemokines, and cellular signaling pathways that contribute to the regulation of immunosenescence and age-related chronic inflammation. These molecules and pathways play critical roles in the progression of chronic inflammation and immunosenescence, which can further impair immune function and heighten the susceptibility to age-related chronic diseases.

#### 2.2.1. Interleukin-1 Family

IL-1, including IL-1α and IL-1β, is an important cytokine initiating the stress-induced inflammatory cascade [24]. It is also a key mediator of immune and inflammatory responses [28]. The pro-inflammatory effects of IL-1 are partly attributable to its ability to activate immune cells and induce the production of other pro-inflammatory cytokines, such as TNF-α and IL-6. Furthermore, IL-1 is involved in the regulating of cell death and survival. It can induce the death of certain cell types, such as pancreatic beta cells, while promoting the survival of others, such as neutrophils. Given its central role in immune and inflammatory responses, IL-1 has been intensively researched in recent years. Strategies to modulate IL-1 signaling are actively being pursued as potential therapeutic approaches for various inflammatory diseases.

#### 2.2.2. Interleukin-6

IL-6, a multifunctional and pleiotropic cytokine, assumes a critical role in regulating the acute-phase response, immunity transition and the pathogenesis of many chronic diseases [25,29]. It is secreted in response to stimulation triggered by systemic inflammation [28]. IL-6 has long been acknowledged as a significant cytokine in age-related diseases and it exhibits context-dependent pro-inflammatory or anti-inflammatory properties via distinct signaling pathways [30]. It has been called the ‘gerontologist’s cytokine’ since 1993 [21]. The levels of IL-6 are generally low in the blood, and serum IL-6 levels are usually undetectable in the absence of inflammation. However, IL-6 can be detected with advancing age, and its levels increase in subjects with markers of frailty and chronic diseases; thus, elevated IL-6 levels are closely correlated with mortality [31,32]. An increased IL-6 expression level is a distinctive feature of the ageing process, potentially indicating the development of age-associated pathological processes that evolve gradually over several decades, even in individuals who appear to be in good health [33,34].

#### 2.2.3. Tumor Necrosis Factor Alpha

TNF-α is a key regulator of the inflammatory response [35]. During the inflammatory process, TNF-α is up-regulated with age [22]. While TNF-α can act locally in tissues as a beneficial pro-inflammatory mediator, its systemic release can be significantly detrimental. Intracellular ageing studies have shown that TNF-α is up-regulated in older individuals, and in octogenarians and centenarians with atherosclerosis [26,36,37]. It has also been reported that high levels of TNF-α production in the supernatants of whole blood samples are associated with a markedly elevated risk of death from cardiovascular events in patients with a long lifespan [38]. These findings suggest that TNF-α exhibits distinct biological effects and serves as a marker of frailty in older individuals. Genetic studies have revealed no differences in the distribution of TNF-α -308 G/A genotypes across centenarians, octogenarians, and younger groups; nevertheless, the G/A genotype was found to be associated with a decreased prevalence of dementia in centenarians [18,39,40]. Moreover, an allele of the TNF-α gene-308 G/A variants is associated with an increased susceptibility to myocardial infarction. TNF-α variants and TNF-α itself have also been linked to varying degrees with an augmented risk of Alzheimer’s disease [41]. Recently, increasing evidence has suggested that dysregulation of TNF-α signaling is associated with age-related diseases, highlighting the importance of developing anti-inflammatory drugs. Research has reported that TNF-α inhibitors may be potential prophylactic or ameliorative agents in age-related diseases [42].

#### 2.2.4. Nuclear Factor-κB (NF-κB) System

NF-κB is a master transcription factor that plays an important role in recognition signaling and host responses to immune challenge. It has been recognized as the most important factor underlying inflammation [23,43]. In the absence of stimulation, inhibitory proteins sequester NF-κB complexes within the cytoplasm. However, upon stimulation, the inhibitory proteins are degraded, allowing the NF-κB complex to translocate to the nucleus; once in the nucleus, it triggers the transcription of several genes, with a particular emphasis on pro-inflammatory genes [44]. Previous studies have shown that the age-related dysregulation of the NF-κB signaling pathway up-regulates the expression of pro-inflammatory cytokines [20,45]. Helenius et al. [27] used the electrophoretic mobility shift assay to examine DNA-binding activities in nuclear extracts obtained from different tissues of young and aged rodents. Their findings revealed a notable elevation in the nuclear levels of NF-κB components, specifically p52 and p65, in aged rodent tissues. These results suggest an enhanced activation of the NF-κB pathway during the process of aging. In addition, in a study by Adler et al. [46], motif mapping was conducted on the promoters of genes that were up-regulated during aging, revealing that the NF-κB transcription factor exhibited the strongest association with the aging process. This age-related persistent activation of NF-κB has been confirmed by multiple researchers who have investigated alterations in the NF-κB pathway across different tissues during ageing [47]. Moreover, Yamashita et al. [48] reported that the NF-κB pathway directly enhances osteoclast differentiation and maturation in periodontitis.

## 3. Periodontitis and Inflammation

Periodontitis, characterized as a chronic inflammatory disease, is induced by bacterial infection, which stems from the complex interactions between the subgingival microbiota and the host immune response. With the persistence of inflammation, the gingiva, periodontal ligament, cementum, and alveolar bone, which constitute the periodontal tissues, become progressively damaged. Extensive destruction of the alveolar bone compromises tooth support, ultimately leading to tooth loss [49]. Periodontitis is a complex inflammatory disease that is induced by dysbiotic microbiota in the subgingival plaque. The accumulation of bacteria in dental plaques can trigger a strong local inflammatory response [50,51]. In this response, there is a release of pro-inflammatory cytokines, chemokines and matrix metalloproteinases, which can lead to tissue damage and bone resorption in the periodontium. In 1976, Page et al. [52] first described the pathogenesis of human periodontitis by proposing the ‘host response hypothesis’. Since then, our comprehension of the pathogenesis of periodontitis has advanced, uncovering the complex interplay among the subgingival biofilm, the host immunoinflammatory response, and subsequent disruptions in bone and connective tissue homeostasis [53,54]. There is a complex interplay between periodontal microbes and the immune system, involving both innate immune responders and adaptive immunity components, including B and T lymphocytes. These immune cells release pro-inflammatory molecules (e.g., IL-1, IL-6, and TNF-α) and enzymes (e.g., collagenases and matrix metalloproteinases) [5].

### 3.1. Bacteria in Periodontitis

The initiation, progression, and recurrence of periodontitis are significantly influenced by the oral microbiota. Although approximately 700 bacterial species exist in the oral cavity, only a small subset, referred to as periodontal bacteria, possess the capability to cause periodontal inflammation and subsequent bone and tissue damage [55]. Among all of the periodontal bacteria, *Porphyromonas gingivalis* (*P. gingivalis*), *Tannerella forsythia* (*T. forsythia*) and *Treponema denticola* (*T. denticola*) are known as the ‘red complexes’ and are particularly pathogenic [56]. In particular, *P. gingivalis* is considered a ‘keystone pathogen’ in chronic periodontitis [57]. Several studies have shown that *P. gingivalis* and *T. forsythia* are strongly linked with chronic periodontitis [58]. Moreover, a study showed that the oral administration of *P. gingivalis* had an impact on the quantity of osteoclasts and osteoblasts [59]. Similarly, alveolar bone resorption was also observed after the oral administration of *Aggregatibacter actinomycetemcomitans* (*A. actinomycetemcomitans*) [60]. *A. actinomycetemcomitans* is linked to localized aggressive periodontitis, where it can serve as a keystone pathogen and is strongly indicative of bone loss in individuals who are susceptible to this condition [61].

#### 3.1.1. Lipopolysaccharide (LPS)

LPS is considered the primary causative agent responsible for the extensive destruction of deep periodontal tissue and periodontal disease [62,63]. LPS is typically comprised of three domains: lipid A, a short-core oligosaccharide, and the O-antigen [64]. Lipid A, also referred to as endotoxin, represents the bioactive portion of LPS and is recognized by the innate immune system [65]. Upon the recognition of LPS, the innate immune system initiates an immune response aimed at eliminating bacterial intruders [66]. LPS can directly induce osteoclastogenesis and bone resorption in vitro and in vivo in mice and rats. Numerous studies have shown that the direct application of LPS to periodontal tissues induces alveolar bone resorption by osteoclasts, resembling the bone loss observed in periodontitis [67,68]. Jiang et al. [69] showed that LPS derived from *Escherichia coli* and *P. gingivalis* prompted the generation of osteoclasts from mouse leukocytes. Furthermore, LPS derived from *E. coli*- and *A. actinomycetemcomitans*-enhanced receptor activator of nuclear factor kappa-Β ligand (RANKL) expression in periodontal ligament cells (PDLCs), and the addition of LPS increased the count of osteoclasts formed [70,71]. In vitro osteoclastogenesis assays often use LPS derived from bacterial species such as *E. coli*, *P. gingivalis* and *A. actinomycetemcomitans*. To better reflect the pathogenesis of periodontitis and alveolar bone destruction, LPS derived from periodontal bacteria have been increasingly used in in vivo and in vitro models [72,73].

#### 3.1.2. Peptidoglycan (PGN)

PGN, the primary cell wall component of gram-positive bacteria, can stimulate the production of inflammatory cytokine [74]. PGN activates the host’s innate immune system and induces the release of cytokines, which trigger off the activation of osteoclasts [75,76]. Takenori et al. [77] showed that PGN from *Actinomyces naeslundii*-induced inflammatory cytokine production and activated osteoclast formation in co-cultures of mouse bone marrow cells and stromal PA6 cells. However, the effects of PGN on osteoclast differentiation can depend on the bacterial species and other factors. Takami et al. [78] discovered that PGN derived from *Staphylococcus aureus* inhibited the initial stage of osteoclast differentiation when RANKL was present. Meanwhile, Jiang et al. [79] showed that PGN from *S. aureus* alone did not induce osteoclast formation. These findings suggest that PGN has similar effects to LPS on osteoclast formation and may play a vital role in the pathogenesis of periodontitis.

#### 3.1.3. Gingipains

Gingipains, which are cysteine proteinases of *P. gingivalis*, are a major group of virulence factors secreted by this periodontopathogenic bacterium. Gingipains are the products of three genes, two of which code for an arginine-specific proteinase (RgpA and RgpB) and one for a lysine-specific proteinase (Kgp) [80]. Previous studies have shown that serum IgG antibodies in patients with periodontitis are primarily reactive to the hemagglutinin/adhesin region of Kgp and RgpA [81,82]. Additionally, Kimito et al. [83] showed that among the antigens produced by *P. gingivalis*, serum IgG levels in patients with periodontitis were the highest against recombinant RgpA. Furthermore, the N-terminus of recombinant RgpA was found to be the most suitable antigen for screening patients with periodontitis. Gingipains also cause immune response dysregulation and inflammation and promote the interactions between *P. gingivalis* and other periodontal pathogens, facilitating their survival and biofilm formation [84,85]. Furthermore, gingipains play a key role in immune regulation by endowing *P. gingivalis* with the ability to evade host immune responses and immune clearance [86]. Moreover, gingipains are involved in the progression of bone loss in periodontitis. In a co-culture system in which gingipains were added to osteoblasts and osteoclast progenitor cells for an osteoclastogenesis assay, the results showed that treatment with Rgp did not alter the number of osteoclasts, but treatment with Kgp promoted osteoclastogenesis [87].

## 4. Periodontitis and Ageing

Periodontitis is a chronic inflammatory disease that affects the underlying supporting alveolar bone and soft tissue surrounding the teeth [88,89,90]. Its pathophysiology has been characterized at the molecular level, and it ultimately triggers the activation of host-derived proteinases that enable marginal periodontal ligament fiber loss and apical migration of the junctional epithelium, and allow for the apical spread of the bacterial biofilm along the root surface [91]. In the 1999 International Workshop on Classification of Periodontal Diseases, researchers recommended the unique features of different periodontitis phenotypes and recognized four different forms of periodontitis [92]. Over the past two decades, researchers, clinicians, and epidemiologists have updated the definition of periodontal disease and provided a more refined understanding of periodontal disease staging and grading. The World Symposium 2017 proposed a new classification system for periodontitis that considers both the severity and complexity of periodontitis management. This new system includes four stages of periodontitis, ranging from stage I (initial periodontitis) to stage IV (advanced periodontitis), based on the extent and severity of periodontal tissue damage and the complexity of the treatment required [91]. Severe periodontitis is a highly prevalent chronic disease that has affected approximately 750 million people worldwide over the past three decades, making it the sixth most prevalent chronic disease among the general population [93]. If left untreated, periodontitis can lead to the continual destruction of tooth-supporting tissues, eventually resulting in tooth loss and a consequent decrease in oral function. Finally, it can affect people’s ability to chew, leading to nutritional deficiencies and potential systemic health consequences [3].

Ageing is a time-dependent process that is characterized by the progressive impairment of biological processes in living organisms [5,6]. Studies have shown that ageing is a major risk factor for most chronic diseases and that these diseases restrict the quality of life, independence, and prosperity of older adults [94]. Age-related multimorbidity is a major health concern for older adults, with more than 70% of the people aged over 65 years suffering from two or more chronic disorders such as diabetes, cancer, heart disease, periodontitis, and stroke, accounting for 66.66% of all deaths each year [95]. Periodontitis affects approximately 50% of the world’s adult population, and its prevalence and severity increase with age [7,8,96]. The risk of developing periodontitis increases at approximately 30–40 years of age and is exacerbated in most adults aged over 50 years [5,9]. Among adults aged over 65 years, periodontitis stands as the primary factor behind tooth loss, resulting in significant detriments to masticatory function, aesthetics, and overall quality of life [9].

### 4.1. Cellular Senescence and PDLCs

Cellular senescence is a fundamental phenomenon characterized by irreversible growth arrest, which can be induced by replicative exhaustion or a range of stressors, such as DNA damage, oxidative stress, and inflammation [97]. The induction of the senescent phenotype can be triggered by DNA-damaging stimuli, such as ionizing radiation, oxidative stress, and inflammation [98,99,100]. Senescent cells can release pro-inflammatory cytokines and growth factors, thereby contributing to chronic inflammation and tissue damage in the ageing body. p16^INK4A^, p53^INK4B^, p21^CIP1^ and β-galactosidase are some common biomarkers of senescent cells and are widely used to identify senescent cells [101]. Senescent cells are mainly found at sites of age-related pathologies [102]. In periodontitis, senescent cells accumulate chronologically in the alveolar bone and contribute to age-related alveolar bone deterioration [103,104]. PDLCs, which are a type of mesenchymal stem cells isolated from the periodontal ligament, play an important role in supporting the collagenous fibers in the dense connective tissue of the tooth and in maintaining periodontal homeostasis and replenishing damaged cells during the healing of dental injuries [105]. Moreover, PDLCs constitute a heterogeneous group of cells exhibiting different stages of differentiation and lineage commitment and include periodontal ligament stem cells (PDLSCs), PDLCs and periodontal ligament fibroblasts [106,107]. Recent studies have shown that PDLCs and PDLSCs are similar in terms of their surface marker expression, multipotent differentiation, and regeneration capabilities. These findings suggest that PDLCs may be analogous to PDLSCs and have the potential to differentiate into various cell types [108]. However, the viability and osteogenic differentiation potential of PDLCs have been shown to decline with age [105]. Aged PDLCs exhibit diminished viability and reduced osteogenic differentiation capacity, potentially contributing to the development, and progression of age-related periodontal diseases.

### 4.2. Senescence-Associated Secretory Phenotype (SASP) in Periodontitis

The SASP is characterized by a formidable and intricate group of pro-inflammatory cytokines, chemokines, and proteases secreted by senescent cells. This enigmatic group of molecules collectively orchestrates a profound and dynamic alteration of the local environment, leaving an indelible mark on the surrounding tissue [109,110]. In clinical settings, various inflammatory cytokines and chemokines are identified in the gingival tissue from patients with periodontitis. These biomarkers are indicative of the host’s dysregulated immune response to the subgingival biofilm and are involved in the pathogenesis of periodontitis. IL-1β, IL-6, TNF-α and IFN-γ have been identified as key players in the pathogenesis of periodontitis. These cytokines are produced by immune cells in response to the subgingival biofilm and can cause tissue damage and bone resorption in the periodontium [111]. As previously mentioned, the level of IL-1β increases in patients with periodontitis compared to its level in healthy subjects [112]. Epithelial cells, lymphocytes, and macrophages are the primary sources of IL-6 secretion, triggered by bacterial LPS, IL-1, and TNF-α, and it stimulates osteoclast formation in vitro [113]. TNF-α enhances the expression of IL-1β, IL-6, and RANKL; however, its level in the gingival crevicular fluid (GCF) does not considerably differ before and after periodontitis treatment [114].

In periodontitis, IL-8 and monocyte chemoattractant protein-1 (MCP-1) attract neutrophils and other leucocytes to the inflammation site. IL-8 is produced by macrophages and epithelial cells in response to the presence or stimulation of IL-1β, TNF-α, and LPS [115,116]. Studies have shown that IL-8 enhances osteoclast differentiation and activity, and recruits polymorphonuclear neutrophils to the inflammation site [117]. Furthermore, elevated IL-8 levels have been observed in the GCF of patients diagnosed with chronic periodontitis compared with periodontally healthy control sites [118]. Macrophages, epithelial cells, and T cells secrete MCP-1 in response to bacterial components, including inflammatory mediators or LPS [119]. The MCP-1 level in the GCF decreases after periodontal treatment, compared with levels during periodontitis (at the affected sites), indicating a reduction in the inflammatory response [120,121].

### 4.3. Alveolar Bone Loss in Periodontitis

Alveolar bone loss is a hallmark of periodontitis. The alveolar bone, recognized as a dynamic and highly regulated tissue, assumes a crucial function in providing support to the teeth and maintaining their proper position within the maxillofacial skeleton [122]. Alveolar bone remodeling is a complex process that is influenced by a range of mechanical, nutritional, and hormonal factors. The homeostasis between bone formation and resorption is critical for maintaining the structural integrity and function of the alveolar bone. This homeostasis is regulated by a range of hormones and cytokines, which coordinate the coupled process of bone formation and resorption to maintain the alveolar bone volume in healthy persons [123,124,125]. However, in periodontitis, the disruption of homeostasis results in the progressive loss of alveolar bone. The incidence of periodontitis rises with ageing. Although aging itself does not directly cause periodontitis, it can exert an influence on the periodontal milieu, potentially affecting the process of bone resorption and coupling. Ageing is associated with an increase in the production of cytokines that stimulate osteoclastogenesis and inhibit osteoblastic bone formation, leading to an imbalance in bone remodeling [126]. In addition, ageing can lead to changes in hormone levels and nutritional status, which can further exacerbate the imbalance between bone formation and resorption. Ageing-related loss of alveolar bone and periodontal attachment are not necessarily separate processes from periodontitis. Instead, ageing may exacerbate the loss of alveolar bone and periodontal attachment occurring in older adults with periodontitis, leading to more severe disease and poorer outcomes [127,128,129].

Periodontitis is characterized by a perturbation of the delicate balance between bone formation and resorption. The immune responses facilitated by periodontal tissues can trigger T-cell activation and subsequent immune cell accumulation in periodontal lesions, leading to local inflammation and bone damage caused by osteoclasts [130]. In 2000, Arron et al. [131] introduced the concept of osteoimmunology, which describes the intricate interplay between the immune and skeletal systems. Osteoclasts and osteoblasts play a pivotal role in bone remodeling, and their interactions with immune cells involve the secretion of cytokines as well as direct cell–cell contact [132]. Dominant osteoclast activity is the primary cause of alveolar bone loss. Upon activation, osteoclasts adhere to the bone surface and engage with various hormones, cytokines, and proteases to break down the bone mineral matrix, leading to the loss of alveolar bone tissue [49]. Resident cells like fibroblasts, keratinocytes, and dendritic cells secrete inflammatory cytokines that promote the migration of numerous inflammatory cells, including neutrophils, macrophages, and T/B cells, towards the site of inflammation. These cells progressively infiltrate the deeper layers to the periodontal connective tissue, including the alveolar bone [133,134]. As a result, the stimulation and activation of osteoclasts, coupled with the inhibition of osteoblasts, disrupts the delicate balance between bone resorption and regeneration, ultimately resulting in a decrease in bone volume [135]. The pathogenesis of alveolar bone loss is a complex and multifaceted process, involving a broad array of cellular and molecular interactions. Meanwhile, tissue engineering has advanced to the point where it can offer the potential to restore lost alveolar bone, periodontal ligament, and root cementum. Tissue engineering has opened new avenues for achieving predictable and optimal periodontal tissue regeneration. Understanding the intricacies of the bone remodeling process and the factors that regulate it is critical to developing effective therapies for periodontitis and other bone-related diseases.

## 5. Conclusions

The prevalence and severity of periodontitis increase with age, making it a significant health concern for the ageing population. As human lifespans continue to extend globally, understanding the impact of ageing on physiological and disease pathological has garnered significant attention in recent years. In this review, we have provided an up-to-date summary of the current knowledge of biological ageing and periodontitis, with a focus on the molecular mechanisms underlying pathological ageing at the periodontal level. Firstly, we have synthesized current knowledge on the pro-inflammatory cytokines, chemokines, and cellular signaling pathways that regulate immunosenescence and age-related chronic inflammation. Elucidating these mechanisms enhances our understanding of the relationship between aging and inflammation. Furthermore, we have summarized the key bacteria associated with periodontitis pathogenesis. Identifying these microbes provides critical insights into the etiology of periodontitis. Overall, this review integrates findings across multiple areas, like immunosenescence, oral microbiome changes, and tissue destruction pathways to provide a comprehensive overview of the linkages between aging and periodontitis. In fact, the interplay between aging and inflammation represents a complex process with no clear established causality or directionality yet. Several challenges persist in our current understanding of the biological mechanisms through which aging influences periodontitis. However, it is evident that age-related processes can contribute to the initiation of inflammation, be a consequence of existing inflammation, or involve a bidirectional relationship between the two phenomena. Given these understanding, we propose the pathways linking inflammation and periodontitis, examining how ageing influences the inflammation–periodontitis linkage and immune resistance (Figure 1). Gaining a more comprehensive understanding of the complex interactions between ageing, inflammation, and periodontitis is critical for developing effective interventions to prevent and treat these conditions in older adults. Further studies should build upon this synthesis of current knowledge to deepen our mechanistic comprehension of this major age-related oral health concern.

## Figures and Tables

**Figure 1 bioengineering-10-01274-f001:**
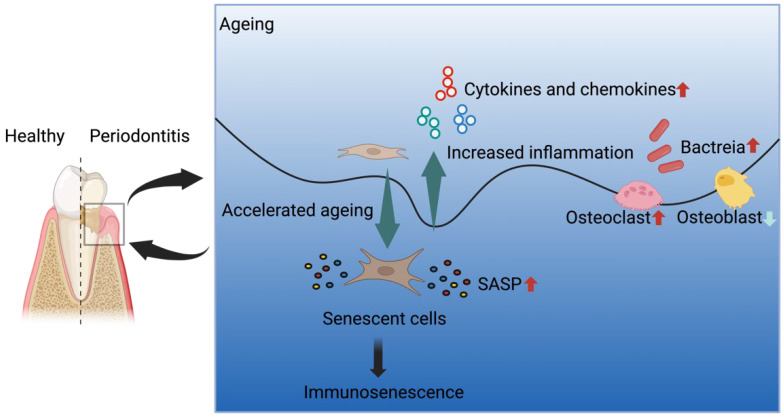
The interplay of factors associated with ageing and inflammation in the development of periodontitis.

**Table 1 bioengineering-10-01274-t001:** The pro-inflammatory cytokines, chemokines, and cellular signaling pathways that contribute to the regulation of immunosenescence and age-related chronic inflammation (↑↑↑ is strong increase, ↑↑ is medium increase, ↑ is mild increase, - is no different).

Cytokines/Chemokines/ Cellular Signalling Pathways	Immunosenescence	Age-Related Chronic Inflammation
IL-1α	↑↑↑	↑
IL-1β	↑↑	↑↑
IL-6	↑↑↑	↑
TNF-α	-	↑
IL-8	↑↑↑	↑
CXCL10	↑↑↑	-
IL-12	↑	↑↑
IFN-α	-	↑
iNOS	-	↑↑↑
GSK3-β	↑	↑
MAPK	↑	↑
PKB	↑	↑↑↑
NF-κB	↑	↑↑↑
References	[19,21,22,23]	[24,25,26,27]

## Data Availability

Not applicable.

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
