# Peer review of "Ageing and Inflammation: What Happens in Periodontium?"

_bioengineering, 2023, doi:10.3390/bioengineering10111274_

Round 1

Reviewer 1 Report

Comments and Suggestions for Authors

In this manuscript, authors reviewed the numerous factors that are related to ageing and inflammation, and in particular those that are involved in periodontium onset. 

Some minor issues to correct:

- To easier the reading, when an author is mentioned, it is better to put the corresponging reference number near author name. Per example in page 2, line 61, Franceschi et al., put the reference number 55 after et al. and not at the end of the phrase: Franceschi et al. [55].  The same for references number 52 (line 183), 53 (line 185), 55 (line 189), 60 (line 207), 78 (line 247), 93 (line 284), 149 (line 430).

- Page 1, Introduction, lines 30 and 31): replace the semicolon by a comma after the words "..cancers, atherosclerosis, diabetes,.." 

- Page 3, Table 1, First column, replace the word "Reference" by "References"

- Page 4, item 2.2.2, line 130: When it say: "multifunctional, pleiotropic..." Replace the comma by "and"

- Page 4, item 2.2.3, lines 162,163: Nowadays it is better to say "variant" instead "polymorphisms"

- Page 7, item 4, line 323: Please revised the references mentioned: "...diseases [4,5,106](120-122). Studies..."

- Page 9, item 4.3, lines 406 and 411: Replace the word "co-ordinated" (line 406) and  "co-ordinate" (line 411) by "coordinated" and "coordinate" respectively.

Author Response

Response: Thank you very much for your great suggestions, which are very helpful to our manuscript. Strictly following your comments, we have carefully checked our manuscript and have revised the sentences accordingly. Thank you very much.

Reviewer 2 Report

Comments and Suggestions for Authors

The manuscript " Ageing and inflammation: what happens in periodontium? "  is original and properly written. The topic is described in detail.  I have only a minor suggestion for the authors: the conclusion should be expanded.

Author Response

Response: Thank you so much for your constructive suggestion. Following your suggestion, we have expanded our conclusion and have made our manuscript clearer. The revised conclusion as follow: The prevalence and severity of periodontitis increase with age, making it a significant health concern for the growing elderly population. As human lifespan continues extending globally, understanding the impact of ageing on physiological and disease pathological has garnered significant attention in recent years. In this review, we have provided an up-to-date summary of the current knowledge of biological ageing and periodontitis, with a focus on the molecular mechanisms underlying pathological ageing at the periodontal level. Firstly, we have synthesized current knowledge on the pro-inflammatory cytokines, chemokines and cellular signalling pathways that regulate immunosenescence and age-related chronic inflammation. Elucidating these mechanisms enhances our understanding of the relationship between aging and inflammation. Furthermore, we have summarized the key bacteria associated with periodontitis pathogenesis. Identifying these microbes provides critical insights into the etiology of periodontitis. Overall, this review integrates findings across multiple are-as—immunosenescence, oral microbiome changes, and tissue destruction pathways—to provide a comprehensive overview of the linkages between aging and periodontitis. Given these understanding, we proposed the pathways linking inflammation and periodontitis, examining how ageing influences the inflammation–periodontitis linkage and immune resistance (Fig. 1). Gaining a more comprehensive the complex interactions between ageing, inflammation and periodontitis is critical for developing effective interventions to prevent and treat these conditions in older adults. Further studies should build upon this synthesis of current knowledge to deepen our mechanistic comprehension of this major age-related oral health concern. Thank you very much.

Reviewer 3 Report

Comments and Suggestions for Authors

Dear Authors, 

you made a great work! However, some improvements are suggested before acceptance. 

Author Response

Response: Thank you so much for your helpful suggestions.

  • As for the keywords, we have added “inflamm-ageing” to improve future research of our manuscript.
  • Regarding to the introduction, thanks for your recommended articles. The first article showed that increased periodontal destruction, the presence of periodontal bacteria and viruses with the increased level of placental mir155 in preeclamptic women with chronic periodontitis. Meanwhile, periodontal pathogens further strengthen the evidence of periodontal inflammation as a risk of some diseases especially when associated with chronic periodontitis. It is relevant to the topic of our review. Therefore, we have cited the reference to make our manuscript more comprehensive. This mentioned reference has been properly cited in the revised manuscript as follow: Meanwhile, the presence of periodontal bacteria and viruses increased periodontal destruction [2].
  • For the second article, it demonstrated that the amount of early marginal bone remodeling is not correlated with periodontitis, but bacteria effected the bone loss in the dental implant. It provided more understanding of cytokines at the periodontal level. For this, we have cited the reference as follow: Meanwhile, bacteria worsen the bone resorption around a dental implant [11]. Thank you very much again.

Reviewer 4 Report

Comments and Suggestions for Authors

This study seeks to clarify recent scientific advancements concerning the role of inflamm-ageing in the deterioration of periodontal function and the development of periodontitis. It aims to accentuate existing research findings and identify potential directions for future investigations in this crucial research domain.

Despite the extensive compilation of literature findings, this paper falls short in conducting an in-depth analysis and establishing connections between inflammation, ageing, and periodontitis. A notable illustration of this deficiency is observed in Figure 1, which regrettably does not offer any substantive insights into this intricate relationship.

While I commend the authors for their diligent work in assembling the evidence, I encourage them to take the final step and provide a more profound understanding of this complex interplay.

Round 2

Reviewer 4 Report

Comments and Suggestions for Authors

The updated version of the manuscript incorporated mostly minor revisions throughout the text, with major alterations made to the conclusion. However, these changes do not effectively fulfill the intended purpose of elucidating recent scientific advancements regarding the role of inflamm-ageing in the decline of periodontal function and the development of periodontitis.

I genuinely appreciate the substantial effort made by the authors to synthesize the existing evidence on this topic. Nevertheless, the paper falls short of delivering substantial clarity or significant insights. I would strongly recommend that the authors consider reworking Figure 1 to provide readers with a graphical representation that aligns with the paper's stated objective of clarification.
